# AMORTIZED CONTEXT VECTOR INFERENCE FOR SEQUENCE-TO-SEQUENCE NETWORKS

## ABSTRACT

Neural attention (NA) has become a key component of sequence-to-sequence models that yield state-of-the-art performance in as hard tasks as abstractive document summarization (ADS), machine translation (MT), and video captioning (VC). NA mechanisms perform inference of context vectors; these constitute weighted sums of *deterministic* input sequence *encodings*, adaptively sourced over long temporal horizons. Inspired from recent work in the field of amortized variational inference (AVI), in this work we consider treating the *context vectors* generated by soft-attention (SA) models as latent variables, with approximate *finite mixture model* posteriors inferred via AVI. We posit that this formulation may yield stronger generalization capacity, in line with the outcomes of existing applications of AVI to deep networks. To illustrate our method, we implement it and experimentally evaluate it considering challenging ADS, VC, and MT benchmarks. This way, we exhibit its improved effectiveness over state-of-the-art alternatives.

## 1 INTRODUCTION

Sequence-to-sequence (*seq2seq*) or encoder-decoder models (Sutskever et al., 2014) constitute a novel solution to inferring relations between sequences of different lengths. They are broadly used for addressing tasks including machine translation (MT) (Bahdanau et al., 2015; Luong et al., 2015), abstractive document summarization (ADS), descriptive caption generation (DCG) (Xu et al., 2016), and question answering (QA) (Sukhbaatar et al., 2015), to name just a few. *Seq2seq* models comprise two distinct RNN models: an *encoder* RNN, and a *decoder* RNN. Their main principle of operation is based on the idea of learning to infer an intermediate *context vector representation, $c$,* which is "shared" among the two RNN modules of the model, i.e., the encoder and the decoder. Specifically, the encoder converts the source sequence to a context vector (e.g., the final state of the encoder RNN), while the decoder is presented with the inferred context vector to produce the target sequence.

Despite these merits, though, baseline *seq2seq* models cannot learn temporal dynamics over long horizons. This is due to the fact that a single context vector $c$ is capable of encoding rather limited temporal information. This major limitation has been addressed via the development of neural attention (NA) mechanisms (Bahdanau et al., 2015). NA has been a major breakthrough in Deep Learning for Natural Language Processing, as it enables the decoder modules of *seq2seq* models to adaptively focus on temporally-varying subsets of the source sequence. This capacity, in turn, enables flexibly capturing long temporal dynamics in a computationally efficient manner.

Among the large collection of recently devised NA variants, the vast majority build upon the concept of *Soft Attention* (SA) (Xu et al., 2016). Under this rationale, at each sequence generation (decoding) step, NA-obtained context vectors essentially constitute deterministic representations of the dynamics between the source sequence and the decodings obtained thus far. However, recent work in the field of amortized variational inference (AVI) (Jimenez Rezende & Mohamed, 2015; Kingma & Welling, 2013) has shown that it is often useful to treat representations generated by deep networks as *latent random* variables. Indeed, it is now well-understood that, under such an inferential setup, the trained deep learning models become more effective in inferring representations that offer stronger generalizaton capacity, instead of getting trapped to representations of poor generalizaton quality. Then, model training reduces to inferring posterior distributions over the introduced latent variables. This can be performed by resorting to variational inference (Attias, 2000), where the sought variational posteriors are parameterized via appropriate deep networks.

Motivated from these research advances, in this paper we consider a novel formulation of SA. Specifically, we propose an NA mechanism formulation where the generated context vectors are considered random latent variables with *finite mixture model posteriors*, over which AVI is performed. We dub our approach amortized context vector inference (ACVI). To exhibit the efficacy of ACVI, we implement it into: (i) Pointer-Generator Networks (See et al., 2017), which constitute a state-of-the-art approach for addressing ADS tasks; (ii) baseline *seq2seq* models with additive SA, applied to the task of VC; and (iii) baseline *seq2seq* models with multiplicative SA, applied to MT.

The remainder of this paper is organized as follows: In Section 2, we briefly present the *seq2seq* model variants in the context of which we implement our method and exhibit its efficacy. In Section 3, we introduce the proposed approach, and elaborate on its training and inference algorithms. In Section 4, we perform an extensive experimental evaluation of our approach using benchmark ADS, MT, and VC datasets. Finally, in the concluding Section, we summarize the contribution of this work.

## 2 METHODOLOGICAL BACKGROUND

### 2.1 ABSTRACTIVE DOCUMENT SUMMARIZATION

ADS consists in not only copying from an original document, but also learning to generate new sentences or novel words during the summarization process. The introduction of *seq2seq* models has rendered ADS both feasible and effective (Rush et al., 2015; Zeng et al., 2017). Dealing with out-of-vocabulary (OOV) words was one of the main difficulties that early ADS models were confronted with. Word and/or phrase repetition was a second issue. The *pointer-generator* model presented in (See et al., 2017) constitutes one of the most comprehensive efforts towards ameliorating these issues.

In a nutshell, this model comprises one bidirectional LSTM (Hochreiter & Schmidhuber, 1997) (BiLSTM) encoder, and a unidirectional LSTM decoder, which incorporates an SA mechanism (Bahdanau et al., 2015). The word embedding of each token, $\boldsymbol{x}_i$, $i \in \{1, \dots, N\}$, in the source sequence (document) is presented to the encoder BiLSTM; this obtains a representation (encoding) $\boldsymbol{h}_i = [\overrightarrow{\boldsymbol{h}}_i; \overleftarrow{\boldsymbol{h}}_i]$, where $\overrightarrow{\boldsymbol{h}}_i$ is the corresponding forward LSTM state, and $\overleftarrow{\boldsymbol{h}}_i$ is the corresponding backward LSTM state. Then, at each generation step, $t$, the decoder LSTM gets as input the (word embedding of the) previous token in the target sequence. During training, this is the previous word in the available reference summary; during inference, this is the previous generated word. On this basis, the decoder updates its internal state, $\boldsymbol{s}_t$, which is then presented to the postulated SA network. Specifically, the attention distribution, $\boldsymbol{a}_t$, is given by:

$$e_t^i = \boldsymbol{v}^T \tanh(\boldsymbol{W}_h \boldsymbol{h}_i + \boldsymbol{W}_s \boldsymbol{s}_t + \boldsymbol{b}_{attn}) \tag{1}$$

$$\boldsymbol{a}_t = \text{softmax}(\boldsymbol{e}_t), \ \boldsymbol{e}_t = [e_t^i]_i \tag{2}$$

where the $\boldsymbol{W}_{\cdot}$ are trainable weight matrices, $\boldsymbol{b}_{attn}$ is a trainable bias vector, and $\boldsymbol{v}$ is a trainable parameter vector of the same size as $\boldsymbol{b}_{attn}$. Then, the model updates the maintained context vector, $\boldsymbol{c}_t$, by taking an weighted average of all the source token encodings; in that average, the used weights are the inferred attention probabilities. We obtain:

$$\boldsymbol{c}_t = \sum_i a_t^i \boldsymbol{h}_i \tag{3}$$

Eventually, the *predictive distribution* over the next *generated* word yields:

$$P_t^{vocab} = \text{softmax}(\boldsymbol{V}' \tanh(\boldsymbol{V}[\boldsymbol{s}_t; \boldsymbol{c}_t] + \boldsymbol{b}) + \boldsymbol{b}') \tag{4}$$

where $\boldsymbol{V}$ and $\boldsymbol{V}'$ are trainable weight matrices, while $\boldsymbol{b}$ and $\boldsymbol{b}'$ are trainable bias vectors.

In parallel, the network also computes an additional probability, $p_t^{gen}$, which expresses whether the next output should be *generated* by sampling from the predictive distribution, $P_t^{vocab}$, or the model should simply *copy* one of the already available s*ource sequence tokens*. This mechanism allows for the model to cope with OOV words; it is defined via a simple sigmoid layer of the form:

$$p_t^{gen} = \sigma(\boldsymbol{w}_c^T \boldsymbol{c}_t + \boldsymbol{w}_s^T \boldsymbol{s}_t + \boldsymbol{w}_x^T \boldsymbol{x}_t + \boldsymbol{b}_{ptr}) \tag{5}$$

where $\boldsymbol{x}_t$ is the decoder input, while the $\boldsymbol{w}_{\cdot}$ and $\boldsymbol{b}_{ptr}$ are trainable parameter vectors. The probability of copying the $i$th source sequence token is considered equal to the corresponding attention probability,

$a_t^i$. Eventually, the obtained probability that the next output word will be $\beta$ (found either in the vocabulary or among the source sequence tokens) yields:

$$P_t(\beta) = p_t^{gen} P_t^{vocab}(\beta) + (1 - p_t^{gen}) \sum_{i:\beta_i=\beta} a_t^i \tag{6}$$

Finally, a *coverage mechanism* may also be employed (Tu et al., 2016), as a means of penalizing words that have already received attention in the past, to prevent repetition. Specifically, the coverage vector, $\boldsymbol{k}_t$, is defined as:

$$\boldsymbol{k}_t = [k_t^i]_{i=1}^N = \sum_{\tau=0}^{t-1} \boldsymbol{a}_\tau \tag{7}$$

Using the so-obtained coverage vector, expression (1) is modified as follows:

$$e_t^i = \boldsymbol{v}^T \tanh(\boldsymbol{W}_h \boldsymbol{h}_i + \boldsymbol{W}_s \boldsymbol{s}_t + \boldsymbol{w}_k k_t^i + \boldsymbol{b}_{attn}) \tag{8}$$

where $\boldsymbol{w}_k$ is a trainable parameter vector of size similar to $\boldsymbol{v}$. Model training is performed via minimization of the categorical cross-entropy, augmented with a coverage term of the form:

$$\lambda \sum_i \sum_t \min(a_t^i, c_t^i) \tag{9}$$

Here, $\lambda$ controls the influence of the coverage term; in the remainder of this work, we set $\lambda = 1$.

## 2.2 VIDEO CAPTIONING

*Seq2seq* models with attention have been successfully applied to several datasets of multimodal nature. Video captioning constitutes a popular such application. In this work, we consider a simple *seq2seq* model with additive SA that comprises a BiLSTM encoder, an LSTM decoder, and an output distribution of the form (4). The used encoder is presented with visual features obtained from a *pretrained* convolutional neural network (CNN). Using a pretrained CNN as our employed visual feature extractor ensures that all the evaluated attention models are presented with identical feature descriptors of the available raw data. Hence, it facilitates fairness in the comparative evaluation of our proposed attention mechanism. We elaborate on the specific model configuration in Section 4.2.

## 2.3 MACHINE TRANSLATION

Machine translation constitutes one of the first sequential data modeling applications where *seq2seq* models were shown to obtain state-of-the-art performance. In this work, we perform MT by means of a baseline *seq2seq* model comprising a BiLSTM encoder, an LSTM decoder, a *predictive distribution* over the next *generated* word which is given by (4), and a multiplicative SA mechanism. The latter is described by (Luong et al., 2015):

$$e_t^i = \boldsymbol{h}_i \boldsymbol{W} \boldsymbol{s}_t \tag{10}$$

in conjunction with Eq. (2); therein, $\boldsymbol{W}$ is a trainable weights matrix. Our consideration here of multiplicative SA both serves the purpose of implementing and evaluating our approach under diverse SA variants, and is congruent with the best reported results in the related literature.

## 3 PROPOSED APPROACH

We begin by introducing the core assumption that the computed context vectors, $\boldsymbol{c}_t$, constitute latent random variables. Further, we assume that, at each decoding step, $t$, the corresponding context vector, $\boldsymbol{c}_t$, is drawn from a distribution associated with one of the available source sequence encodings, $\{\boldsymbol{h}_i\}_{i=1}^N$. The selection of the source sequence encoding to associate with is determined from the output sequence via the decoder state, $\boldsymbol{s}_t$, as we explain next.

Let us introduce the set of binary latent indicator variables, $\{z_t^i\}_{i=1}^N$, $z_t^i \in \{0, 1\}$, with $z_t^i = 1$ denoting that the context vector $\boldsymbol{c}_t$ is drawn from the $i$th density, that is the density associated with the $i$th source encoding, $\boldsymbol{h}_i$, and $z_t^i = 0$ otherwise. Then, we postulate the following hierarchical model:

$$\boldsymbol{c}_t | z_t^i = 1; \mathcal{D} \sim p(\boldsymbol{\theta}(\boldsymbol{h}_i)) \tag{11}$$

$$z_t^i = 1 | \mathcal{D} \sim \pi_t^i(a_t^i) \tag{12}$$

where $\mathcal{D}$ comprises the set of *both the source and target* training sequences, $\boldsymbol{\theta}$ denotes the parameters set of the context vector conditional density, and $\pi_t^i$ denotes the probability of drawing from the $i$th conditional at time $t$. Notably, we assume that the component assignment probabilities, $\pi_t^i$, are functions of the attention probabilities, $a_t^i$. Thus, the selection of the mixture component density that we draw the context vector from at decoding time $t$ is directly determined from the value of the current decoder state, $\boldsymbol{s}_t$, via the corresponding attention probabilities. A higher affinity of the current decoder state $\boldsymbol{s}_t$ with the $i$th encoding, $\boldsymbol{h}_i$, at time $t$, results in higher probability that the context vector be drawn from the corresponding conditional density.

Having defined the hierarchical model (11)-(12), it is important that we examine the resulting expression of the posterior density $p(\boldsymbol{c}_t; \mathcal{D})$. By marginalizing over (11) and (12), we obtain:

$$p(\boldsymbol{c}_t; \mathcal{D}) = \sum_{i=1}^{N} \pi_t^i(a_t^i) \, p(\boldsymbol{\theta}(\boldsymbol{h}_i)) \tag{13}$$

In other words, we obtain a *finite mixture model posterior over the context vectors,* with mixture conditional densities associated with the available source sequence encodings, and mixture weights that are functions of the corresponding attention vectors, and are therefore determined by the target sequences.

In addition, it is interesting to compare this expression to the definition of context vectors under the conventional SA scheme. From (3), we observe that conventional SA is merely a special case of our proposed model, obtained by introducing two assumptions: (i) that the postulated mixture component assignment probabilities are identity functions of the associated attention probabilities, i.e.

$$p(\boldsymbol{z}_t; \mathcal{D}) = \text{Cat}(\boldsymbol{z}_t | \boldsymbol{\pi}_t), \, \boldsymbol{z}_t = [z_t^i]_{i=1}^N, \, \boldsymbol{\pi}_t = [\pi_t^i(a_t^i)]_{i=1}^N$$
$$\text{s.t.} \quad \pi_t^i(a_t^i) \triangleq p(z_t^i = 1; \mathcal{D}) = a_t^i = \text{softmax}(\boldsymbol{e}_t); \tag{14}$$

and (ii) that the conditional densities of the context vectors have all their mass concentrated on $\boldsymbol{h}_i$, that is they collapse onto the single point, $\boldsymbol{h}_i$:

$$p(\boldsymbol{c}_t | z_t^i = 1; \mathcal{D}) = \delta(\boldsymbol{h}_i) \tag{15}$$

Indeed, by combining (13) - (15), we yield:

$$p(\boldsymbol{c}_t; \mathcal{D}) = \sum_{i=1}^{N} a_t^i \delta(\boldsymbol{h}_i) \tag{16}$$

whence we obtain (3) with probability 1.

Thus, our approach replaces the simplistic conditional density expression (15) with a more appropriate family $p(\boldsymbol{\theta}(\boldsymbol{h}_i))$, as in (13). Based on the literature of AVI, e.g. (Jimenez Rezende & Mohamed, 2015; Kingma & Welling, 2013; Sønderby et al., 2016), we posit that such a *stochastic* latent variable consideration may result in significant advantages for the postulated *seq2seq* model. Specifically, our trained model becomes more agile in searching for effective context representations, as opposed to getting trapped to poor local solutions.

In the following, we examine conditional densities of Gaussian form. Adopting the inferential rationale of AVI, we consider that these conditional Gaussians are parameterized via the postulated BiLSTM encoder. Specifically, we assume:

$$p(\boldsymbol{c}_t | z_t^i = 1; \mathcal{D}) = \mathcal{N}\big(\boldsymbol{c}_t | \boldsymbol{h}_i, \text{diag}(\boldsymbol{\sigma}^2(\boldsymbol{h}_i))\big) \tag{17}$$

where

$$\log \boldsymbol{\sigma}^2(\boldsymbol{h}) = \text{ReLU}(\boldsymbol{h}) \tag{18}$$

$\text{ReLU}(\cdot)$ is a trainable ReLU layer of size $\dim(\boldsymbol{h})$, and the encodings, $\boldsymbol{h}_i$, are obtained from a BiLSTM encoder, similar to conventional models. Hence:

$$p(\boldsymbol{c}_t; \mathcal{D}) = \sum_{i=1}^{N} a_t^i \mathcal{N}\big(\boldsymbol{c}_t | \boldsymbol{h}_i, \text{diag}(\boldsymbol{\sigma}^2(\boldsymbol{h}_i))\big) \tag{19}$$

Thus, we have arrived at an approximate (variational) posterior expression for the context vectors, $c_t$. In our variational treatment, both the component-conditional means, $h_i$, and their variances, $\sigma^2(h_i)$, are obtained from (amortizing) neural networks presented with the source sequences. On the other hand, though, the assignment probabilities, $\pi_t^i$, in the variational posterior are taken as the attention probabilities, $a_t^i$. Thus, they are determined by the target sequences, which are generated from the decoder of the model. Hence, our treatment represents a valid approximate posterior formulation, overall conditioned on both the source and target sequences.

This concludes the formulation of ACVI.

**Relation to Recent Work.** From the above exhibition, it becomes apparent that our approach generalizes the concept of neural attention by introducing stochasticity in the computation of context vectors. As we have already discussed, the ultimate goal of this construction is to allow for inferring representations of better generalization capacity, by leveraging Bayesian inference arguments.

We emphasize that this is in stark contrast to recent efforts toward generalizing neural attention by deriving more complex attention distributions. For instance, (Kim et al., 2017) have recently introduced *structured attention*. In that work, the model infers complex posterior probabilities over the assignment latent variables, as opposed to using a simplistic gating function. Specifically, instead of considering independent assignments, they postulate the first-order Markov dynamics assumption:

$$p(\{z_t\}_{t=1}^T; \mathcal{D}) = p(z_1; \mathcal{D}) \prod_{t=1}^{T-1} p(z_{t+1}|z_t; \mathcal{D}) \tag{20}$$

Thus, (Kim et al., 2017) compute posterior distributions over the attention assignments, while ACVI provides a method for obtaining improved representations through the inferred context vectors. Note also that Eq. (20) gives rise to the need of executing much more computationally complex algorithms to perform attention distribution inference, e.g. the forward-backward algorithm (Rabiner, 1989). In contrast, our method imposes computational costs comparable to conventional SA.

Similar is the innovation in the *variational attention* method, recently presented (Deng et al., 2018). In essence, its key conceptual difference from *structured attention* is the consideration of full independence between the attention assignments $\{z_t\}_{t=1}^T$. Among the several alternatives considered in (Deng et al., 2018) to obtain stochastic gradient estimators of low variance, it was found that an approach using REINFORCE (Williams, 1992) along with a specialized baseline was effective.

Another noteworthy recent work, closer related to ACVI, is the variational encoder-decoder (VED) method presented in (Bahuleyan et al., 2018). Among the several alternative formulations considered in that paper, the one that clearly outperformed the baselines in terms of the obtained accuracy (BLEU scores) combined *seq2seq* models with SA with an extra variational autoencoder (VAE) module. This way, apart from the context vector, which is computed under the standard SA scheme, an additional latent vector $\xi$ is essentially inferred. The imposed prior over it is a standard $\mathcal{N}(0, I)$, while the inferred posterior is a diagonal Gaussian parameterised by a BiLSTM network presented with the input sequence; the final BiLSTM state vector is presented to dense layers that output the posterior means and variances of the latent vectors $\xi$. Both the context vector, $c$, as well as the latent vectors, $\xi$, are fed to the final softmax layer of the model that yields the generated output symbols.

We shall provide comparisons to all these related approaches in the experimental section of our paper.

**Training Algorithm.** To perform training of a *seq2seq* model equipped with the ACVI mechanism, we resort to maximization of the resulting evidence lower-bound (ELBO) expression. To this end, we need first to introduce some prior assumption over the context latent variables, $c_t$. To serve the purpose of simplicity, and also offer a valid way to effect model regularization, we consider:

$$p(c_t) = \mathcal{N}(c_t|0, I) \tag{21}$$

On the grounds of these assumptions, it is easy to show that the resulting ELBO expression becomes:

$$\mathcal{L} = \sum_t \left\{ \mathbb{E}_{p(c_t; \mathcal{D})}[-J_t] - \text{KL}[p(c_t; \mathcal{D})||p(c_t)] \right\} \tag{22}$$

In this expression, $\mathbb{E}_{p(c_t; \mathcal{D})}[-J_t]$ is the posterior expectation of the model log-likelihood, which is an integral part of the ELBO definition. In the following, we approximate all the entailed ELBO

Table 1: Abstractive Document Summarization: Scores on the test set.

| Method | ROUGE | | | METEOR | |
|---|---|---|---|---|---|
| | 1 | 2 | L | Exact Match | + stem/syn/para |
| *seq2seq* with SA | 31.33 | 11.81 | 28.83 | 12.03 | 13.20 |
| pointer-generator + coverage: **SA** | 39.53 | 17.28 | 36.38 | 17.32 | 18.72 |
| transformer | 24.40 | 5.89 | 17.60 | 10.38 | 10.72 |
| pointer-generator + coverage: **structured attention** | 40.12 | 17.61 | 36.74 | 17.38 | 18.93 |
| pointer-generator + coverage: **variational attention** | 40.04 | 17.37 | 36.45 | 17.14 | 18.66 |
| pointer-generator + coverage: **VED** | 41.28 | 18.05 | 38.12 | 17.63 | 18.87 |
| pointer-generator + coverage: **ACVI** | **42.71** | **19.24** | **39.05** | **18.47** | **20.09** |

terms by drawing MC samples from the context vector posterior. In this work, we are dealing with a one-out-of-many predictive selection; hence, the model likelihood is a simple Categorical. As such, $\mathbb{E}_{p(\boldsymbol{c}_t; \mathcal{D})}[-J_t]$ essentially reduces to the negative categorical cross-entropy of the model, averaged over multiple MC samples of the context vectors, drawn from (19). Besides, to ensure that the resulting MC estimators will be of low variance, we adopt the reparameterization trick. To this end, we rely on the posterior expressions (17) and (14); we express the drawn MC samples as follows:

$$\boldsymbol{c}_t^{(k)} = \sum_{i=1}^{N} z_{ti}^{(k)} \boldsymbol{c}_{ti}^{(k)} \tag{23}$$

In this expression, the $\boldsymbol{c}_{ti}^{(k)}$ are samples from the conditional Gaussians (17), which employ the standard reparameterization trick rationale, as applied to Gaussian variables:

$$\boldsymbol{c}_{ti}^{(k)} = \boldsymbol{h}_i + \boldsymbol{\sigma}(\boldsymbol{h}_i) \circ \boldsymbol{\epsilon}_{ti}^{(k)}, \ \boldsymbol{\epsilon} \sim \mathcal{N}(\boldsymbol{0}, \boldsymbol{I}) \tag{24}$$

On the other hand, the $z_{ti}^{(k)}$ are samples from the Categorical distribution (14). To allow for performing backpropagation through these samples, while ensuring that the obtained gradients will be of low variance, we may draw $z_{ti}^{(k)}$ by making use of the Gumbel-Softmax relaxation (Jang et al., 2017). We have empirically found it suffices that we employ the Gumbel-Softmax trick for the last 10% of the model training iterations[1]; previously, we merely adopt the following heuristic, without any statistically significant performance deviation: We use a simple weighted average of the samples $\boldsymbol{c}_{ti}^{(k)}$, with the weights being the attention probabilities, $a_t^i$:

$$\boldsymbol{c}_t^{(k)} \leftarrow \sum_{i=1}^{N} a_t^i \boldsymbol{c}_{ti}^{(k)} \tag{25}$$

This way, we alleviate the computational costs of employing the Gumbel-Softmax relaxation, which dominates the costs of sampling from the mixture posterior (19).

Having obtained a reparameterization of the model ELBO that guarantees low variance estimators, we proceed to its maximization by resorting to a modern, off-the-shelf, stochastic gradient optimizer. Specifically, we adopt simple stochastic gradient descent (SGD) for the MT tasks, and Adam with its default settings (Kingma & Ba, 2015) for the rest.

**Inference Algorithm.** To perform target decoding by means of a *seq2seq* model that employs the ACVI mechanism, we resort to *Beam search* (Russel & Norvig). In our experiments, *Beam width* is set to five for the ADS and VC tasks (Sections 4.1 and 4.2), and to ten for the MT tasks (Section 4.3).

# 4 EXPERIMENTAL EVALUATION[2]

## 4.1 ABSTRACTIVE DOCUMENT SUMMARIZATION

Our experiments are based on the *non-anonymized CNN/Daily Mail* dataset, similar to the experiments of (See et al., 2017). To obtain some comparative results, we use pointer-generator networks as our

---

[1] We use a Gumbel-Softmax temperature hyperparameter value of 0.5.

[2] We have developed our source codes in Python, using the TensorFlow library (Abadi et al., 2015).

Table 2: Abstractive Document Summarization: Novel words generation rate and OOV words adoption rate obtained by using pointer-generator networks.

|  | SA | Structured Attention | Variational Attention | VED | ACVI |
|---|---|---|---|---|---|
| Rate of Novel Words | 0.05 | 0.05 | 0.05 | 0.12 | 0.38 |
| Rate of OOV Words Adoption | 1.16 | 1.18 | 1.18 | 1.21 | 1.25 |

Table 3: Video Captioning: Performance of the considered alternatives.

| Method | ROUGE: Valid. Set | ROUGE: Test Set | CIDEr: Valid. Set | CIDEr: Test Set |
|---|---|---|---|---|
| SA | 0.5628 | 0.5701 | 0.4575 | 0.421 |
| Structured Attention | 0.5804 | 0.5712 | 0.5071 | 0.4283 |
| Variational Attention | 0.5809 | 0.5716 | 0.5103 | 0.4289 |
| VED | 0.5839 | 0.5749 | 0.5421 | 0.4298 |
| ACVI | **0.5968** | **0.5766** | **0.6039** | **0.4375** |

evaluation platform (See et al., 2017); therein, we employ our ACVI mechanism, the standard SA mechanism used in (See et al., 2017), VED (Bahuleyan et al., 2018), variational attention (Deng et al., 2018), as well as structured attention using the first-order Markov assumption (20) (Kim et al., 2017). The observations presented to the encoder modules constitute 128-dimensional word embeddings of the original 50K-dimensional one-hot-vectors of the source tokens. Similarly, the observations presented to the decoder modules are 128-dimensional word embeddings pertaining to the summary tokens (reference tokens during training; generated tokens during inference). Both these embeddings are trained, as part of the overall training procedure of the evaluated models. To allow for faster training convergence, we split training into five phases, as suggested in (See et al., 2017). Following the suggestions in (See et al., 2017), we evaluate all approaches with LSTMs that comprise 256-dimensional states and do not employ Dropout. We have tested VED with various selections of the dimensionality of the autoencoder latent vectors, $\xi$; we report results with 128-dimensional latent vectors, which yielded the best performance in our experiments[3].

Finally, for completeness sake, we also evaluate the Transformer network (Vaswani et al., 2017), which is a popular alternative to *seq2seq* models with SA, based on the notion of self-attention. Following Fevry (2018), Transformer is evaluated with 256-dimensional word embeddings, 4 encoding and decoding layers of 256 units each, 4 heads, and a Dropout rate of 0.2.

We use ROUGE[4] (Lin, 2004) and METEOR[5] (Denkowski & Lavie, 2014) as our performance metrics. METEOR is evaluated both in exact match mode (rewarding only exact matches between words) and full mode (additionally rewarding matching stems, synonyms and paraphrases). In all our experiments, we restrict the used vocabulary to the 50K most common words in the considered dataset, similar to (See et al., 2017). Note that this is significantly smaller than typical in the literature (Nallapati et al., 2016). Our quantitative evaluation is provided in Table 1. Some indicative examples of generated summaries can be found in Appendix A (Tables 7-10).

As we observe, utilization of ACVI outperforms all the alternatives by a large margin. It is also interesting that the Transformer network yields the lowest performance among the considered alternatives; the obtained results are actually very poor. This is commensurate with the results reported by other researchers, e.g. Fevry (2018).

Finally, it is interesting to examine whether ACVI increases the propensity of a trained model towards generating *novel words,* that is words that *are not found* in the source document, as well as the capacity to adopt OOV words. The related results are provided in Table 2. We observe that ACVI increases the number of generated novel words by 3 times compared to the best performing alternative, that is VED (Bahuleyan et al., 2018). In a similar vein, ACVI appears to help the model better cope with OOV words.

---

[3]This selection is also commensurate with the $\xi$ dimensionality reported in Bahuleyan et al. (2018).

[4]`pypi.python.org/pypi/pyrouge/`.

[5]`www.cs.cmu.edu/~alavie/METEOR`.

Table 4: Translation results on the (En, Vi) and (En, Ro) pairs.

| | | BLEU | | | | | | | |
|---|---|---|---|---|---|---|---|---|---|
| Source→Target Language | | En→Vi | | Vi→En | | En→Ro | | Ro→En | |
| Method | | dev | test | dev | test | dev | test | dev | test |
| | Baseline | 23.21 | 25.18 | 20.89 | 23.28 | 12.87 | 14.40 | 15.87 | 15.78 |
| | Structured Attention | 23.81 | 25.00 | 21.19 | 23.08 | 14.04 | 15.08 | 17.02 | 17.68 |
| | Variational Attention | 23.48 | 25.54 | 21.13 | 23.61 | 14.02 | 15.51 | 17.49 | 17.40 |
| | VED | **24.47** | 25.31 | 21.32 | 23.80 | 12.84 | 12.76 | 15.18 | 15.56 |
| | ACVI | 24.08 | **26.16** | 21.26 | **24.47** | **14.15** | **15.78** | **18.07** | 17.78 |
| | Transformer | 24.34 | 25.68 | **21.40** | 23.92 | 13.90 | 15.30 | 17.66 | **17.91** |

## 4.2 VIDEO CAPTIONING

Our evaluation of the proposed approach in the context of a VC application is based on the Youtube2Text video corpus (Yao et al., 2015). We split the available dataset into a training set comprising the first 1,200 video clips, a validation set composed of 100 clips, and a test set comprising the last 600 clips in the dataset. To reduce the entailed memory requirements, we process only the first 240 frames of each video. To obtain some initial video frame descriptors, we employ a pretrained GoogLeNet CNN (Szegedy et al., 2015) (implementation provided in Caffe (Jia et al., 2014)). Specifically, we use the features extracted at the *pool5/7x7_s1 layer* of this pretrained model. We select 24 equally-spaced frames out of the first 240 from each video, and feed them into the prescribed CNN to obtain a 1024 dimensional frame-wise feature vector. These are the visual inputs presented to the trained models. All employed LSTMs entail 1000-dimensional states. These are mapped to 100-dimensional features via the matrices $W_h$ and $W_s$ in Eq. (1). The autoencoder latent variables, $\xi$, of VED are also selected to be 100-dimensional vectors. The decoders are presented with 256-dimensional word embeddings, obtained in a fashion similar to our ADS experiments. In all cases, we use Dropout with a rate of 0.5.

We yield some comparative results by evaluating *seq2seq* models configured as described in Section 2.2; we use ACVI, structured attention in the form (20), VED, variational attention, or the conventional SA mechanism. Our quantitative evaluation is performed on the grounds of the ROUGE-L and CIDEr (Vedantam et al., 2015) scores, on both the validation set and the test set. The obtained results are depicted in Table 3; they show that our method outperforms the alternatives by an important margin. It is also characteristic that Structured Attention yields essentially identical results with Variational Attention. Thus, the first-order Markovian assumption does not offer practical benefits when generating short sequences like the ones involved in VC. Finally, we provide some indicative examples of the generated results in Appendix B (Figs. 1-8).

## 4.3 MACHINE TRANSLATION

Our experiments make use of publicly available corpora, namely WMT'16 English-to-Romanian (En→Ro) and Romanian-to-English (Ro→En), as well as IWSLT'15 English-to-Vietnamese (En→Vi) and Vietnamese-to-English (Vi→En). We benchmark the evaluated models against word-based vocabularies, and present our results in terms of the BLEU score (Papineni et al., 2002).

Following the related literature, we utilize *byte pair encoding* (BPE) (Sennrich et al., 2016) in the case of the (En, Ro) pair. This allows for seamlessly handling rare words, by breaking a given vocabulary into a fixed-size vocabulary of variable-length character sequences (subwords). Subword vocabularies are shared among the languages of a source/destination pair. This way, we promote frequent subword units, thus improving the coverage of the available dictionary words.

We obtain some comparative performance results by evaluating *seq2seq* models using ACVI, conventional SA, structured attention, as well as both the variational alternatives (Bahuleyan et al., 2018; Deng et al., 2018) discussed in Section 3. The trained architecture is homogeneous across all our comparisons. Specifically, both the encoders and the decoders of the evaluated models are presented with 256-dimensional *trainable* word embeddings. We utilize 2-layer BiLSTM encoders, and 2-layer LSTM decoders; all comprise 256-dimensional hidden states on each layer, similar to the summarization task, and employ a Dropout rate of 0.2. For VED, we employ 100-dimensional latent variables $\xi$, following Bahuleyan et al. (2018). Finally, we also provide the performance of the

Table 5: Abstractive Document Summarization: Domain Adaptation Performance on DUC2004.

| Model | ROUGE-1 | ROUGE-2 | ROUGE-L |
|---|---|---|---|
| SA | 27.02 | 7.44 | 22.69 |
| Variational Attention | 27.65 | 7.58 | 23.50 |
| VED | 30.68 | 9.97 | 27.02 |
| ACVI | 32.09 | 10.88 | 28.14 |

Transformer network. The trained Transformer network comprises 4 heads, 4 encoder/decoder layers of 256 units, and a Dropout rate of 0.1.

Our results in Table 4 show inferior performance for our variational inference-based competitors. We observe that VED is competitive to ACVI in two of the four development sets, but fails to generalize as well across test sets. Some indicative examples of generated outputs from the considered variational alternatives are provided in Appendix C, Tables 11-16.

### 4.4    FURTHER INVESTIGATION: DOMAIN ADAPTATION

Finally, we wish to examine the capability of ACVI to generalize across domains. We have already elaborated on our expectation that modeling the context vectors as latent random variables should yield improved generalization performance. We attribute to this fact the improved accuracy ACVI obtained in our experimental evaluations. However, if this is the case, one would probably expect the method to also generalize better across different domains.

To investigate this aspect, we use the trained ADS models described in Section 4.1 to generate summaries for the documents of the DUC2004 dataset [6]. This is an English dataset comprising 500 documents. Each document contains 4 model summaries written by experts. In Table 5, we show how our method performs in this setting, and how it compares to the alternative variational methods considered in Section 4.1. We observe that ACVI yields a clear improvement over the alternatives, while all variational methods perform significantly better than baseline SA. These findings seem to support our theoretical intuitions.

## 5    CONCLUSIONS

In this work, we cast the problem of context vector computation for *seq2seq*-type models employing SA into amortized variational inference. We made this possible by considering that the sought context vectors are latent variables following a Gaussian mixture posterior; therein, the mixture component densities depend on the source sequence encodings, while the mixture weights depend on the target sequence attention probabilities. We exhibited the merits of our approach on *seq2seq* architectures addressing ADS, VC, and MT tasks; we used benchmark datasets in all cases.

We underline that our approach induces only negligible computational overheads compared to conventional SA. Specifically, the only extra trainable parameters that our approach postulates stem from Eq. (17); these are of extremely limited size compared to the overall model size, and correspond to merely few extra feedforward computations at inference time. Besides, our sampling strategy does not induce significant computational costs, since we adopt the reparameterization (25) for the most part of the model training algorithm. In the future, we aim to consider how ACVI can cope with power-law distributions (Chatzis & Demiris, 2012; Chatzis & Kosmopoulos, 2015); such a capacity is of importance to real-world natural language generation.

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

Table 6: Abstractive Document Summarization: Training phases.

| Phase | Iterations | Max encoding steps | Max decoding steps |
|---|---|---|---|
| 1 | 0 - 71k | 10 | 10 |
| 2 | 71k - 116k | 50 | 50 |
| 3 | 116k - 184k | 100 | 50 |
| 4 | 184k - 223k | 200 | 50 |
| 5 | 223k - 250k | 400 | 100 |

## APPENDIX A

We provide some further details on the experimental setup of Section 4.1. To begin with, the used dataset comprises 287,226 training pairs of documents and reference summaries, 13,368 validation pairs, and 11,490 test pairs. In this dataset, the average article length is 781 tokens; the average summary length is 3.75 sentences, with the average summary being 56 tokens long.

To allow for faster training convergence, we split it into five phases, following See et al. (2017). On each phase, we employ a different number of maximum encoding steps for the evaluated models (i.e., the size of the inferred attention vectors), as well as for the maximum allowed number of decoding steps. We provide the related details in Table 6. During these phases, we train the employed models with the coverage mechanism being disabled; that is, we set $w_k = 0$. We enable this mechanism only after these five training phases conclude. Specifically, we perform a final 3K iterations of model training, during which we train the $w_k$ weights along with the rest of the model parameters.

In Tables 7-10, we provide some indicative examples of produced summaries. We also show what the initial document has been, as well as the available reference summary used for quantitative performance evaluation. In all cases, we annotate OOV words in italics, we highlight novel words in purple, we show contextual understanding in bold, while article fragments also included in the generated summary are highlighted in green.

Table 7: Example 223.

| Article |
| --- |
| lagos , nigeria -lrb- cnn -rrb- a day after winning nigeria 's presidency , muhammadu buhari told cnn 's christiane amanpour that he plans to aggressively fight corruption that has long plagued nigeria and go after the root of the nation 's unrest . buhari said he 'll " rapidly give attention " to curbing violence in the northeast part of nigeria , where the terrorist group boko haram operates . by cooperating with neighboring nations chad , cameroon and niger , he said his administration is confident it will be able to thwart criminals and others contributing to nigeria 's instability . for the first time in nigeria 's history , the opposition defeated the ruling party in democratic elections . buhari defeated incumbent goodluck jonathan by about 2 million votes , according to nigeria 's independent national electoral commission . the win comes after a long history of military rule , coups and botched attempts at democracy in africa 's most populous nation . in an exclusive live interview from abuja , buhari told amanpour he was not concerned about reconciling the nation after a divisive campaign . he said now that he has been elected he will turn his focus to boko haram and " plug holes " in the " corruption infrastructure " in the country . " a new day and a new nigeria are upon us , " buhari said after his win tuesday . " the victory is yours , and the glory is that of our nation . " earlier , jonathan phoned buhari to concede defeat . the outgoing president also offered a written statement to his nation . " i thank all nigerians once again for the great opportunity i was given to lead this country , and assure you that i will continue to do my best at the helm of national affairs until the end of my tenure , " jonathan said . " i promised the country free and fair elections . (...) |
| **Reference Summary** |
| muhammadu buhari tells cnn 's christiane amanpour that he will fight corruption in nigeria . nigeria is the most populous country in africa and is grappling with violent boko haram extremists . nigeria is also africa 's biggest economy , but up to 70 % of nigerians live on less than a dollar a day . |
| **Generated Summary** |
| *muhammadu buhari* talks to cnn 's christiane amanpour **about the nation 's unrest** . for the first time in nigeria , opposition defeated incumbent goodluck jonathan by about 2 million votes. *buhari* : " the victory is yours , and the glory is that of our nation " |

Table 8: Example 89.

| Article |
| --- |
| lrb- cnn -rrb- eyewitness video showing white north charleston police officer michael slager shooting to death an unarmed black man has exposed discrepancies in the reports of the first officers on the scene . slager has been fired and charged with murder in the death of 50-year-old walter scott . a bystander 's cell phone video , which began after an alleged struggle on the ground between slager and scott , shows the five-year police veteran shooting at scott eight times as scott runs away . scott was hit five times . if words were exchanged between the men , they 're are not audible on the tape . it 's unclear what happened before scott ran , or why he ran . the officer initially said that he used a taser on scott , who , slager said , tried to take the weapon . before slager opens fire , the video shows a dark object falling behind scott and hitting the ground . it 's unclear whether that is the taser . (...) |
| **Reference Summary** |
| more questions than answers emerge in controversial s. c. police shooting . officer michael slager , charged with murder , was fired from the north charleston police department . |
| **Generated Summary** |
| video **shows** white north charleston police officer michael *slager* shooting to death . *slager* has been charged with murder in the death of 50-year-old walter scott . the video shows a dark object falling behind scott and hitting the ground . |

Table 9: Example 1305.

| Article |
| --- |
| andy murray came close to giving himself some extra preparation time for his wedding next week before ensuring that he still has unfinished tennis business to attend to . the world no 4 is into the semi-finals of the miami open , but not before getting a scare from 21 year-old austrian dominic thiem , who pushed him to 4-4 in the second set before going down 3-6 6-4 , 6-1 in an hour and three quarters . murray was awaiting the winner from the last eight match between tomas berdych and argentina 's juan monaco . prior to this tournament thiem lost in the second round of a challenger event to soon-to-be new brit aljaz bedene . andy murray pumps his first after defeating dominic thiem to reach the miami open semi finals . muray throws his sweatband into the crowd after completing a 3-6 , 6-4 , 6-1 victory in florida . murray shakes hands with thiem who he described as a ' strong guy ' after the game . (...) |
| **Reference Summary** |
| british no 1 defeated dominic thiem in miami open quarter finals . andy murray celebrated his 500th career win in the previous round . third seed will play the winner of tomas berdych and juan monaco in the semi finals of the atp masters 1000 event in key biscayne |
| **Generated Summary** |
| the world no 4 is into the semi-finals of the miami open . *murray* **is still ahead** of his **career** **through the season** . **andy** *murray* was awaiting the winner from the last eight match . *murray* throws his sweatband into the crowd after a 6-4 6-1 victory in florida . |

Table 10: Example 1710.

| Article |
| --- |
| steve clarke afforded himself a few smiles on the touchline and who could blame him ? this has been a strange old season for reading , who are one win away from an fa cup semi-final against arsenal but have spent too long being too close to a championship relegation battle . at least this win will go some way to easing that load . they made it hard for themselves , but they had an in-form player in jamie mackie who was able to get the job done . he put reading in front in the first half and then scored a brilliant winner just moments after chris o'grady had levelled with a penalty -- one of the only legitimate chances brighton had all night , even if clarke was angry about the decision . reading frontman jamie mackie fires the royals ahead against brighton in tuesday 's championship fixture . mackie -lrb- centre -rrb- is congratulated by nathaniel chalobah and garath mccleary after netting reading 's opener . reading -lrb- 4-1-3-2 -rrb- : federici ; gunter , hector , cooper , chalobah ; akpan ; mcleary , williams -lrb- keown 92 -rrb- , robson-kanu -lrb- pogrebnyak 76 -rrb- ; blackman , mackie -lrb- norwood 79 -rrb- . subs not used : cox , yakubu , andersen , taylor . scorer : mackie , 24 , 56 . booked : mcleary , pogrebnyak . brighton -lrb- 4-3-3 -rrb- : stockdale ; halford , greer , dunk , bennett ; ince -lrb- best 75 -rrb- , kayal , forster-caskey ; ledesma -lrb- bruno 86 -rrb- , o'grady , lualua . subs not used : ankergren , calderon , hughes , holla , teixeira . scorer : o'grady -lrb- pen -rrb- , 53 . booked : ince , dunk , bennett , greer . ref : andy haines . attendance : 14,748 . ratings by riath al-samarrai . (...) |
| **Reference Summary** |
| reading are now 13 points above the championship drop zone . frontman jamie mackie scored twice to earn royals all three points . chris o'grady scored for chris hughton 's brighton from the penalty spot . niall keown - son of sportsmail columnist martin - made reading debut . |
| **Generated Summary** |
| **jamie** *mackie* **opened** **the scoring against brighton in tuesday 's championship fixture** . chris *o'grady* and *garath mccleary* both scored . **jamie** *mackie* **and** *garath mccleary* **were both involved** in the game . |

APPENDIX B

The considered Video Captioning task utilizes a dataset that comprises 1,970 video clips, each associated with multiple natural language descriptions. This results in a total of approximately 80,000 video / description pairs; the used vocabulary comprises approximately 16,000 unique words. The constituent topics cover a wide range of domains, including sports, animals and music. We preprocess the available descriptions only using the *wordpunct tokenizer* from the NLTK toolbox[7].

Moving on, we provide some characteristic examples of generated video descriptions. In the captions of the figures that follow, we annotate minor deviations with blue color, and use red color to indicate major mistakes which imply wrong perception of the scene.

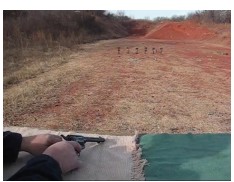 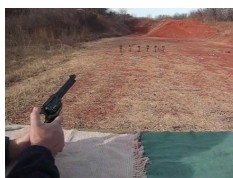 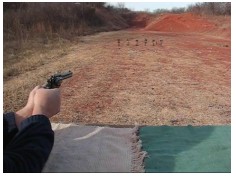 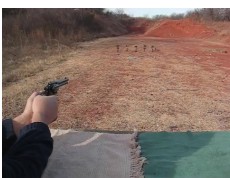

Figure 1: **ACVI:** a man is firing a gun
**VED:** a man is firing a gun
**Structured Attention:** a man is firing a gun
**Variational Attention:** a man is firing a gun
**SA:** a man is firing a gun
**Reference Description:** a man is firing a gun at targets

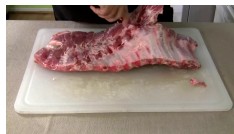 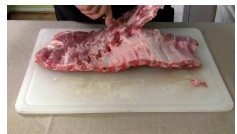 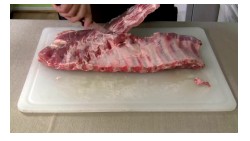 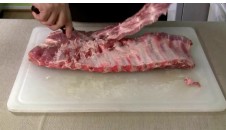

Figure 2: **ACVI:** a woman is cutting a piece of pork
**VED:** a woman is cutting a bed
**Structured Attention:** a woman is cutting pork
**Variational Attention:** a woman is cutting pork
**SA:** a woman is putting butter on a bed
**Reference Description:** someone is cutting a piece of meat

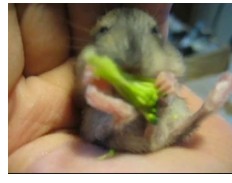 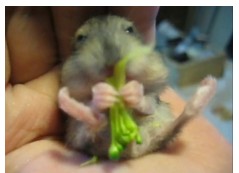 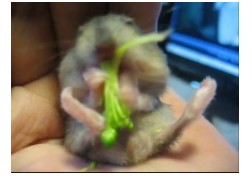 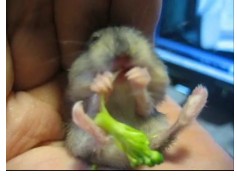

Figure 3: **ACVI:** a small animal is eating
**VED:** a small woman is talking
**Structured Attention:** a small woman is eating
**Variational Attention:** a small woman is eating
**SA:** a small woman is talking
**Reference Description:** a hamster is eating

---

[7]http:/s/www.nltk.org/index.html.

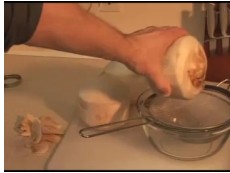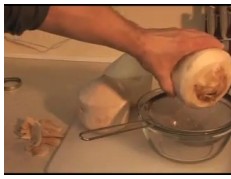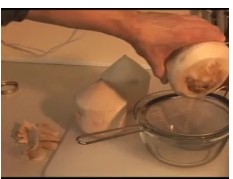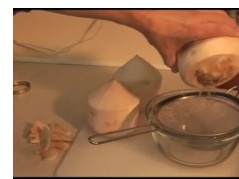

Figure 4: **ACVI:** the lady poured the something into a bowl
**VED:** a woman is cracking an egg
**Structured Attention:** a woman poured an egg into a bowl
**Variational Attention:** a woman poured an egg into a bowl
**SA:** a woman is cracking an egg
**Reference Description:** someone is pouring something into a bowl

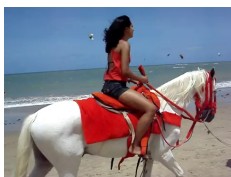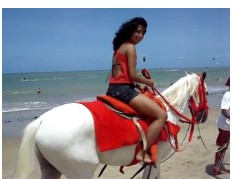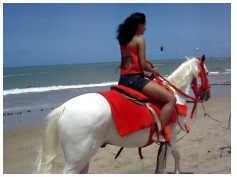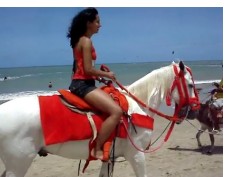

Figure 5: **ACVI:** a woman is riding a horse
**VED:** a woman is riding a horse
**Structured Attention:** a woman is riding a horse
**Variational Attention:** a woman is riding a horse
**SA:** a woman is riding a horse
**Reference Description:** a woman is riding a horse

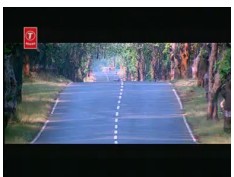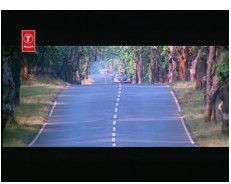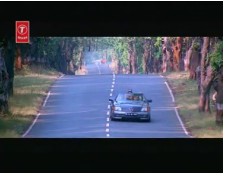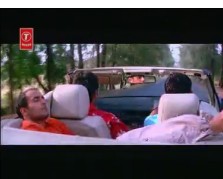

Figure 6: **ACVI:** several people are driving down a street
**VED:** several people trying to jump
**Structured Attention:** several people are driving down the avenue
**Variational Attention:** several people are driving down the avenue
**SA:** a boy trying to jump
**Reference Description:** a car is driving down the road

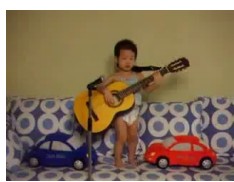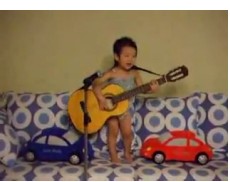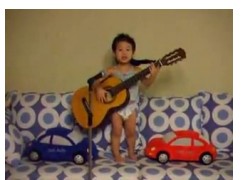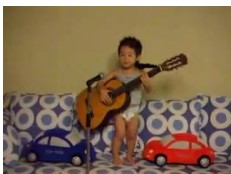

Figure 7: **ACVI:** a man is playing the guitar
**VED:** a man is dancing
**Structured Attention:** a high man is playing the guitar
**Variational Attention:** a man is dancing
**SA:** a high man is dancing
**Reference Description:** a boy is playing the guitar

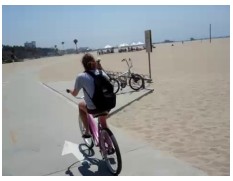 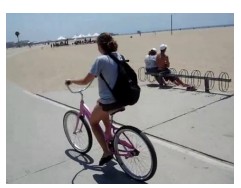 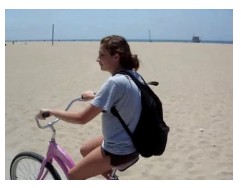 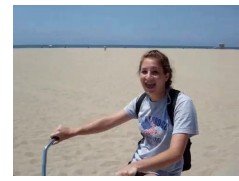

Figure 8: **ACVI:** the man is riding a bicycle
**VED:** the man is riding a motorcycle
**Structured Attention:** the man is riding a motorcycle
**Variational Attention:** the man is riding a motorcycle
**SA:** a man rides a motorcycle
**Reference Description:** a girl is riding a bicycle

Table 11: Vi→En, tst2012 - Example 84.

| Source sentence |
| --- |
| Hầu hết ý tưởng của chúng tôi đều điên khùng, nhưng vài ý tưởng vô cùng tuyệt vời, và chúng tôi tạo ra đột phá. |

| Reference Translation |
| --- |
| Most of our ideas were crazy, but a few were brilliant, and we broke through. |

| Generated Translation - Baseline |
| --- |
| Most of our ideas were crazy, but some incredible ideas were awesome, and we created the breakthrough. |

| Generated Translation - Structured Attention |
| --- |
| Most of our ideas are crazy, but some **[missing: verb]** really wonderful ideas, and we created a sudden. |

| Generated Translation - VED |
| --- |
| Most of our ideas were crazy, but some **[missing: verb]** wonderful ideas, and we created a breakthrough. |

| Generated Translation - Variational Attention |
| --- |
| Most of our ideas were insane, but some ideas were wonderful, and we created <unk>. |

| Generated Translation - ACVI |
| --- |
| Most of our ideas were crazy, but some **[missing: verb]** wonderful ideas, and we made a breakthrough. |

## APPENDIX C

Let us first provide some details on the datasets used in the context of our MT experiments. The WMT'16 task comprises of data from combining the Europarl v7, News Commentary v10 and Common Crawl corpora. For the (En, Ro) pair, this amounts to ~400K parallel sentences. The shared vocabulary sizes (obtained from BPE) total ~31.7K words. We use newsdev2016 as our development set (~1.9K sentences), and newstest2016 as our test set (~1.9K sentences) for the (En, Ro) pair.

On the other hand, the IWSLT'15 task boasts a dataset with ~133K training sentence pairs from translated TED talks, provided by the IWSLT 2015 Evaluation Campaign (Cettolo et al., 2015). Following the same preprocessing steps as in (Luong et al., 2015), we use TED tst2012 (~1.5K sentences) as our validation set for hyperparameter tuning, and TED tst2013 (~1.3K sentences) as our test set. The Vietnamese and English vocabulary sizes are ~7.7K and ~17.2K, respectively.

We prefer default settings for the hyperparameters of the trained *seq2seq* models, as used in the code[8]. These hyper-parameters remain unchanged for the VED and Variational Attention implementations as well. We have migrated the code[9] of the former, provided from the authors, to ensure identical data processing. For the latter, we use their codebase[10] directly.

In conclusion, we provide some characteristic examples of generated translations for all examined models. In the Tables that follow, we annotate minor and major deviations from the reference translation with blue and red respectively. Synonyms are highlighted with green. We also indicate missing tokens by adding the **[missing]** identifier mid-sentence, i.e. verbs, articles, adjectives, etc.

---

[8] https://github.com/harvardnlp/struct-attn.

[9] https://github.com/HareeshBahuleyan/tf-var-attention

[10] https://github.com/harvardnlp/var-attn

Table 12: Vi→En, tst2012 - Example 165.

| Source sentence |
| --- |
| điều đầu tiên bà muốn con hứa là con phải luôn yêu thương mẹ con |
| **Reference Translation** |
| She said, " The first thing I want you to promise me is that you 'll always love your mom. " |
| **Generated Translation - Baseline** |
| " The first thing she wants to revenge is she always loves her mother. " |
| **Generated Translation - Structured Attention** |
| " The first thing she wants to do is always love her. " |
| **Generated Translation - VED** |
| " The first thing she wanted me to do is to love my mother. " |
| **Generated Translation - Variational Attention** |
| " The first thing she wants to promise is that you have to love her mother. " |
| **Generated Translation - ACVI** |
| " The first thing she wanted you to promise you would have to do is to love your mother. " |

Table 13: Vi→En, tst2012 - Example 1542.

| Source sentence |
| --- |
| Họ thậm chí sẽ sử dụng những công cụ như Trojan Scuinst để lây nhiễm vào máy tính của bạn , và từ đó họ có thể có được mọi thông tin bạn trao đổi , có được mọi cuộc hội thoại qua mạng của bạn , và có được mật khẩu của bạn . |
| **Reference Translation** |
| They will even use tools like State Trojan to infect your computer with a trojan , which enables them to watch all your communication , to listen to your online discussions , to collect your passwords . |
| **Generated Translation - Baseline** |
| They're even going to use tools like \<unk\> \<unk\> to infect your computer, and from that they can get all sorts of information that you traded, you get all the conversation through your lives, and there's been available to be able to get all of **[missing: end of sentence]** |
| **Generated Translation - Structured Attention** |
| They're even going to use your tools like \<unk\> \<unk\> to infect your computer, and from that they can get all the information you communicate, there's all kinds of conversations through your network, and you get your password. |
| **Generated Translation - VED** |
| They're even going to use tools like \<unk\> \<unk\> to infect your computer, and then they can get all sorts of information that you share, whether you can get all your \<unk\>. |
| **Generated Translation - Variational Attention** |
| They're even going to use tools like \<unk\> \<unk\> to infect your computer, and then they can be able to get all of the information that you can change, there's your conversation through your online, and there's your password. |
| **Generated Translation - ACVI** |
| They're even going to use tools like \<unk\> \<unk\> to infect your computer, and then they can get all the information you communicate, get all your conversations through your network, and get your password. |

Table 14: Ro→En, newsdev2016 - Example 5.

| Source sentence |
| --- |
| Dirceu este cel mai vechi membru al Partidului Muncitorilor aflat la guvernare luat în custodie pentru legăturile cu această schemă. |
| **Reference Translation** |
| Dirceu is the most senior member of the ruling Workers ' Party to be taken into custody in connection with the scheme. |
| **Generated Translation - Baseline** |
| That is the most old Member of the People 's Party of Maiers to government in custody for ties with this scheme. |
| **Generated Translation - Structured Attention** |
| It is the oldest member of the Mandi of the Massi in the government in the government. |
| **Generated Translation - VED** |
| (RO) Mr President, it is the oldest member of the Dutch Party on the government in custody for the ties with this scheme . |
| **Generated Translation - Variational Attention** |
| It is the oldest Member of the Party of Women's Party of Government in custody for the ties with this scheme. |
| **Generated Translation - ACVI** |
| Dirse is the oldest member of the People 's Party on government in custody for the links with this scheme. |

Table 15: Ro→En, newsdev2016 - Example 7.

| Source sentence |
| --- |
| A fost arestat la începutul lui august de acasă, unde deja se afla sub arest la domiciliu, cu o pedeapsă de 11 ani pentru implicarea într-o schemă de cumpărare a voturilor în Congres cu peste 10 ani în urmă. |
| **Reference Translation** |
| He was arrested early August in his home, where he already was under house arrest serving an 11-year sentence for his involvement in a cash-for-votes scheme in Congress more than 10 years ago. |
| **Generated Translation - Baseline** |
| He was arrested at the beginning of August at home, where it is already under arrest at home, with a death penalty for the involvement of the votes in Congress on 10 years ago. |
| **Generated Translation - Structured Attention** |
| It has been arrested at the beginning of last August, which is already being found in home, with a ban on a 11 years for the involvement of a ban in the reception scheme for more than 10 years ago. |
| **Generated Translation - VED** |
| He was arrested at the beginning of August at home, where it is already under arrest at home, with a three-11 sentence for the involvement in a no-fly scheme on 10 years ago. |
| **Generated Translation - Variational Attention** |
| It was arrested at the beginning of August August, where already under home, with a 11 years **[missing: noun]**, with a 11 years **[missing: noun]** for the involvement of a purchasing votes in 10 years ago. |
| **Generated Translation - ACVI** |
| He was arrested at the beginning of August at home, where he is under house arrest, with a punishment of 11 years for involving a purchasing scheme in Congress over 10 years ago. |

Table 16: Ro→En, newsdev2016 - Example 182.

| Source sentence |
| --- |
| Reprezentanții grupurilor de interese au vorbit la unison despre speranța lor în abilitatea lui Turnbull de a satisface interesul public, de a ajunge la un acord politic și de a face lucrurile bine. |
| **Reference Translation** |
| With one voice the lobbyists talked about a hoped-for ability in Turnbull to make the public argument, to cut the political deal and get tough things done. |
| **Generated Translation - Baseline** |
| The representatives of interest groups have spoken about their hope in the capacity of tourism to meet public interest, to reach a political agreement and to do things well. |
| **Generated Translation - Structured Attention** |
| The representatives of the interest groups have spoken in mind about their hope to meet the public interest, to achieve a political and **good thing**. |
| **Generated Translation - VED** |
| The representatives of interest groups have spoken in unity about their hope in Turkey's ability to satisfy the public interest, to reach a political agreement and to make things right. |
| **Generated Translation - Variational Attention** |
| The representatives of the interest groups have talked about their hope about their hope of their Turk hope to meet the public interest, to reach a political agreement and to do so well. |
| **Generated Translation - ACVI** |
| Representatives of interest groups have spoken about their hope in Mr Turnchl 's ability to satisfy the public interest, to reach a political agreement and to do things well. |

