# OpenReview forum: "Amortized Context Vector Inference for Sequence-to-Sequence Networks"
_ICLR.cc/2019/Conference_

### Official Review · AnonReviewer1 · 2018-11-02
**This paper presents a new way to construct the context vector in seq2seq networks.**

**Rating:** 5
**Confidence:** 4

**Review:**

This paper presents a new way to construct the context vector in seq2seq networks. Specifically, the context vector in the proposed model is treated as a latent variable with a normal prior. The posterior is "defined" as a mixture of the latent presentations of the input sequence and the mixing weight is the attention. To do inference, maximising VAE-like "ELBO" is used. The proposed model is reported to achieve better performance in the seq2seq tasks including document summarisation and video caption.

My comments are as follows:

1. Although the authors claim a full statistical treatment of the context vector, the proposed model is not a proper statistical setting to me. I do not mean the setting is wrong, but it is really weird to me. The paper treats the context vector as a latent variable, which is used to generate the target sentence (take document summarisation task for example). The proper setting is to use the target sentence to do inference of the posterior of the context vector. However, the paper directly defines the format the "posterior", which is a mixture of the hidden representation of the source sentences, ignoring the target sentence. Therefore, I do not think this is the true posterior of the context vector. Moreover, seriously speaking, the "ELBO" of the proposed model is not a real ELBO. It is a training object function looked like an ELBO. Therefore, the difference of the proposed context vector with the standard one is instead of using a fixed hidden representation, the proposed one uses a stochastic one drawn from a Gaussian, the covariance of which is from a neural network.

2. To me, it is less intuitive to analyse why the performance gain comes from, given the proposed context vector works so better than the previous ones. I am expecting the authors to give some intuitive explanations and empirical results on this.

3. Minor comments on clarity: The abstract says both the mean and the covariance are modelled with neural networks, but in the paper, it seems that only the covariance is from a neural network.

---

> ### Author Response · Authors · 2018-11-08
> **Unfortunately the reviewer has missed some core aspects of the postulated posterior and the model ELBO**
>
> 1. The reviewer claims that "The proper setting is to use the target sentence to do inference of the posterior of the context vector."
>
> A.: To be correct (we suppose this is what the reviewer means) the posterior must be conditioned on both the input and the target sequence. That's exactly what we do in this paper. It's such a pity that the reviewer has missed this detail.
> Specifically, the assumed posterior is given in Eq. (13); this is conditioned on \mathcal{D}. We have emphasised below Eq. (12) that \mathcal{D} comprises the set of both source and target training sequences. Why is this the case?
>
> It is true that, in our variational treatment, the component-conditional means, h_i, and variances, σ2(h_i), are obtained from amortizing neural networks presented with the source sequences. On the other hand, though, the component weights π_t^i are taken as the attention probabilities, a_t^i. This way, the so-obtained posteriors depend on both the source sequences, presented to the encoder, as well as the target sequences, that are generated from the decoder of the model; this is obvious from Eqs. (1), (8), and (10).
>
> Hence, our treatment represents a valid approximate posterior formulation, overall conditioned on both the source and target sequences. The source sequences determine the means and variances of the mixture component densities (via the encoder states), the target sequences determine the mixture weights (via the decoder states); both the mixture component densities and the mixture component weights are equally crucial elements of the postulated mixture model posteriors.
>
>
> 2." Therefore, I do not think this is the true posterior of the context vector."
>
> A.: We mays stress that, in all cases, we are not talking about the true posterior, but a variational approximation of valid form.
>
>
> 3. "Moreover, seriously speaking, the "ELBO" of the proposed model is not a real ELBO. It is a training object function looked like an ELBO."
>
> A.: It's not obvious to us how the reviewer supports this claim. The ELBO in variational inference is not something we are not familiar with (we have been working on variational inference since 2006) and we would be grateful to know exactly what the reviewer actually has in mind.
>
> In our provided ELBO expression,  $\mathbb{E}_{p(\boldsymbol{c}_{t};\mathcal{D})}[-J_{t}]$ is the posterior expectation of the model log-likelihood, which is an integral part of the ELBO definition. In a multi-class classification setting, like ours, the model likelihood is a Categorical distribution (over the vocabulary terms).
>
> In our work, we approximate all the entailed ELBO terms by drawing MC samples from the context vector posterior. Then, given our model's Categorical likelihood, $\mathbb{E}_{p(\boldsymbol{c}_{t};\mathcal{D})}[-J_{t}]$ essentially reduces to the negative categorical cross-entropy of the model, averaged over multiple MC samples of the context vectors.
>
> Hence, our ELBO does reduce to the typical objective function (-negative- categorical cross-entropy) averaged over many MC samples, plus the KL term. Our ELBO is correct, and our presentation was intended to facilitate readers not very familiar with variational inference. Unfortunately, it seems it unexpectedly confused the reviewer.
>
> 4. "the proposed one uses a stochastic one drawn from a Gaussian, the covariance of which is from a neural network." -- "The abstract says both the mean and the covariance are modelled with neural networks, but in the paper, it seems that only the covariance is from a neural network."
>
> A.: Let us stress it's not only the covariance but the mean as well. h_t is obtained from the encoder, which is an RNN, i.e. a neural network as well...
>
>
> 5. "To me, it is less intuitive to analyse why the performance gain comes from, given the proposed context vector works so better than the previous ones. I am expecting the authors to give some intuitive explanations and empirical results on this."
>
> A.: We assume the reviewer is seeking some intuitive explanation of why the method works. The search of such intuitive explanations is a common trend, especially among practitioners. Unfortunately, such an answer is not always straightforward, especially when it comes to Deep Learning techniques combined with Bayesian inference arguments.
>
> The theoretical arguments supporting (amortised) variational inference, as opposed to obtaining simple point estimators, are generally known to the community, and we summarise the most pertinent of them in the Introduction (third paragraph), as well as right after Eq. (16). We have extensively elaborated in the paper on our assumption that the reason why our method yields better accuracy is the better generalization performance obtained by modeling the context vectors as latent random variables. These are similar to the arguments employed by other recent related works (which we summarise in the "Relation to Recent Work" subsection). See also Sec. 4.4 in the revised paper.

---

### Official Review · AnonReviewer2 · 2018-11-04
**Amortized Context Vector Inference for Sequence-to-Sequence Networks**

**Rating:** 6
**Confidence:** 3

**Review:**

Summary:
This paper tries to introduce stochasticity over context representation to enable the neural networks to search for effective representation so as to avoid getting trapped into local optima. The basic idea is to formulate the posterior probability of context representation via a mixture gaussian model. The standard attention model SA can be thought of as a special case of the proposed method. The proposed method outperforms a number of baselines on ADS, video captioning, and machine translation datasets.

Strength:
  - Generalize better via considering uncertainty and exploring more search space with negligible computational overheads
  - Outperforms a number of baselines.

Comments:
  - For video captioning task, various methods perform quite similar on test set, but vary a lot on validation set?
  - Any chances to compare bounds of different variational methods from optimization perspective?
  - In order to further validate the generalization capacity, any experiments on low-source data or domain adaptation?
  - Compare with transformer?

---

> ### Author Response · Authors · 2018-11-12
> **Thanks for the feedback**
>
> 1. "For video captioning task, various methods perform quite similar on test set, but vary a lot on validation set? "
> A.: It seems all methods work much better in the validation set than in the test set. We suppose the way the dataset was split, the validation split was much "easier" than the "test" set. If this is the case, one would expect an "easy" dataset to be much easier for a more potent model compared to the alternatives. On the other hand, a "hard" dataset should be more difficult for all models. We suppose this fact has reduced the performance differences among all methods in the test set.
>
> 2. "Any chances to compare bounds of different variational methods from optimization perspective?"
> A.: We are extremely reluctant to introduce in our paper a discussion on how the ELBOs compare across variational inference-based models. From our long experience on variational inference, we know pretty well that the ELBO is a very brittle measure. Seemingly "benign" changes, including different initialization schemes and a different method for reducing MC gradient variance (including different sorts of reparameterization tricks), render totally meaningless any comparison of yielded ELBO values across different models. Effectively, one can run the same variational inference-based model with (minor) changes of these aspects, retain the same accuracy, yet get a significantly different ELBO value.
>
> As nicely summarised in the recent excellent work of:
>
> Yuling Yao, Aki Vehtari, Daniel Simpson and Andrew Gelman, "Yes, but Did It Work?: Evaluating Variational Inference," Proc. ICML 2018
>
> "There are many situations where the VI approximation is flawed. This can be due to the slow convergence of the optimization problem, the inability of the approximation family to capture the true posterior, the asymmetry of the true distribution, the fact that the direction of the KL divergence under-penalizes approximation with too-light tails, or all these reasons. (...) Purely relying on the objective function or the equivalent ELBO does not solve the problem. An unknown multiplicative constant exists in p(θ, y) \propto p(θ | y) that changes with reparametrization, making it meaningless to compare ELBO across two approximations. Moreover, the ELBO is a quantity on an uninterpretable scale"
>
> For these reasons, we opt not to provide such comparisons, as we would be reinforcing a trend which, to our mind, is rather misleading.
>
> 3. " In order to further validate the generalization capacity, any experiments on low-source data or domain adaptation?"
> A.: In the revised version, we will provide a "further investigation" subsection (4.4) where we will use the trained ADS models to summarise the documents in DUC2004. This is a sound domain adaptation experiment.
>
> 4. We shall provide some comparisons to the Transformer network in the revised paper. These will be only dealing with the ADS and MT tasks; we are not aware of a Transformer network variant addressing the multi-modal kind of application that VC is about.

---

### Official Review · AnonReviewer3 · 2018-11-06
**Variational Auto-Encoder**

**Rating:** 6
**Confidence:** 3

**Review:**



The authors made a method that consider latent binary representation instead of continuous representations in Seq2seq models, and showed that it could help regularization in multiple sequence prediction tasks.

To be effective, the authors considered a simple approximation of the average embedding that prevent them from using REINFORCE.

The intuition is that hard latent binary units might be better at representing vocabulary choices and leads to a better inductive bias of the model. However, no inspection of the model to understand in which sentences the averaging of binary variables makes a real difference in prediction.

There is no analysis of the condition under which this approximation is correct. What is the impact of this approximation? Are the results the same if reinforce is used?

Overall, the article is clearly written and conclusions sufficiently relevant for this article to be published in ICLR.

Minor comments:
- the term Amortized Variational Inference is not the most intuitive and Variational Auto-Encoder seems to be the de-facto standard term for these techniques.
- For the MT results, why the authors did not consider more common language pairs so that the results are readily comparable?

---

> ### Author Response · Authors · 2018-11-11
> **Thanks for the feedback**
>
> 1. Let us first focus on the Gumbel-Softmax approximation. Initially, we emphasize that there was an omission in the submitted paper: We used our heuristic (25) for the initial 90% of model training; then, we actually used Gumbel-Softmax with a temperature hyperparameter of 0.5. We will clarify this in the revised paper, at page 6.
>
> In essence, during model training, we need to sample context vectors. Since these follow a Gaussian mixture posterior, we have to first select the component they come from and then draw a sample from that Gaussian component. To allow for backprop to take place, this (hard) selection procedure must be replaced with a relaxation.
>
> Gumbel-Softmax is a provably correct approximation; that is, the relaxed solution converges to the exact one.
> Gumbel-Softmax essentially yields a vector of numbers that take in [0,1] and sum to 1. At the initial stages of training, this vector resembles the mixture weights vector (attention vector) due to the initialization of the temperature hyper-parameter of the relaxation. As training proceeds, the sampling allows for the training procedure to explore the probability space in a stochastic manner. At the final stages, this vector almost becomes a one-hot-vector, resembling a real draw from a Categorical distribution.
>
> At this point, we make a crucial observation: The draws of z (from Gumbel-Softmax) are not used on their own; rather, they are multiplied with samples from the corresponding Gaussians, and then summed. Only the outcomes of this multiplication/summation are eventually used in model training/backprop. Hence, even though replacing z samples with the attention vectors does not facilitate a stochastic exploration (which sampling ensures), this lack may be adequately compensated for by the stochastic nature of the Gaussian samples they are multiplied with.
>
> These observations offered the intuition for our omission of the Gumbel-Softmax for the most part of model training. We have observed no statistically significant differences in BLEU scores between using our scheme or Gumbel-Softmax with conventional temperature annealing schemes. We do not claim the heuristic (25) is correct; we merely have empirically found it's sufficient for reducing computational costs without affecting modeling performance.
>
>
> 2. Regarding the MT datasets, we resorted to these as opposed to larger ones for two main reasons: (i) En-Ro requires BPE while En-Vi does not; hence, these two allow for performing evaluation under both scenarios; (ii) the limited memory capacity of our available hardware did not allow for running experiments on larger datasets without reducing the number of hidden units of the seq2seq architecture and/or the minibatch size; in both cases, our results would have not been directly comparable with published work where bigger architectures are considered, so as to yield the most out of the datasets and seq2seq model capabilities. In contrast, in our provided experiments, model and batch sizes are similar to the literature and the suggestions of the authors of the software packages we have utilised, mainly Tensorflow-NMT. Note also that En-Vi is the standard demo in the Tensorflow-NMT package, that we have largely based our experiments on.
>
>
> 3. Regarding the name of the method, the"auto-encoder" term is often associated with unsupervised (or semi-supervised) learning. Since our method is supervised, end-to-end trainable, we prefer not to use this term to avoid confusion.

---

### Meta-Review · Area_Chair1 · 2018-12-14
**Not quite enough for acceptance**

**Confidence:** 5
**Recommendation:** Reject

**Metareview:**

The overall view of the reviewers is that the paper is not quite good enough as it stands. The reviewers also appreciate the contributions so taking the comments into account and resubmit elsewhere is encouraged.